# On the System-Level Effectiveness of Physical Object-Hiding Adversarial Attack in Autonomous Driving

## Abstract

In Autonomous Driving (AD) systems, perception is crucial for both security and safety. Among the different attacks on AD perception, the physical object-hiding adversarial attacks are especially severe due to their direct impact on road safety. However, we find that all existing works so far only evaluate their attack effect at the targeted AI component level, without any evaluation *at the system level*, i.e., with the entire system semantics and context such as the full AD system pipeline and closed-loop control. This thus inevitably raise a critical research question: can these existing research efforts actually effectively achieve the desired system-level attack effects (e.g., causing vehicle collisions, traffic rule violations, etc.) in the real-world AD system context? In the paper, we perform the first measurement study on whether and how effective the existing designs can lead to system-level effects, where we take the STOP sign-hiding attack as our target. Our evaluation results show that all the representative prior works cannot achieve any system-level effect in a classical closed-loop AD setup in road speeds controlled by common STOP signs. We then point out two limitation hypotheses that appear in all existing works: 1) the unpractical STOP sign size distribution in pixel sampling, and 2) missing particular consideration in system-critical attack range. Our results demonstrate that after overcoming these two limitations, the system-level effects can be further improved, i.e., the violation rate can increase around 70%.

## 1 Introduction

Autonomous Driving (AD) vehicles are now a reality in our daily life, where a wide variety of commercial and private AD vehicles are driving on the road. For example, the millions of Tesla cars (Kane, 2021) are equipped with Autopilot (Tesla, 2022). To ensure safe and correct driving, a fundamental pillar in the AD system is *perception*, which is designed to detect surrounding objects in real time. Due to the direct impact on safety-critical driving decisions such as collision avoidance, various prior works have studied the security of AD perception, especially the ones that aim at causing the disappearance of critical physical road objects (e.g., STOP signs), or *physical object-hiding adversarial attacks* (Jia et al., 2022; Xu et al., 2020; Chen et al., 2018; Wu et al., 2020).

Although a plethora of prior works studied such physical object-hiding adversarial attacks in AD settings, we find that *all* of them only evaluate their attack effect *at the targeted AI component level* (i.e., judged by per-frame object misdetection rates (Chen et al., 2018; Eykholt et al., 2018; Xu et al., 2020; Zhao et al., 2019; Jia et al., 2022)), without any evaluation *at the system level*, i.e., with the full system semantics and context enclosing such AI component (e.g., the remaining AD system pipeline such as object tracking, planning, and control, closed-loop control, and the attack-targeted driving scenario), which we call the *system model* for such adversarial attacks in this paper (§2). This thus inevitably raises a critical research question: can these existing works on physical object-hiding adversarial attacks effectively achieve the desired system-level attack effects (e.g., causing vehicle collisions, traffic rule violations, etc.) in the real-world AD system context?

To systematically answer this critical research question, we take the necessary first step by performing a measurement study on prior works with regard to their capabilities in causing system-level effects. We select STOP sign-hiding attack as our target considering its high representativeness in physical object-hiding adversarial attack today (Shen et al., 2022), and its direct impacts on driving correctness and road safety. We first classify the existing STOP sign-hiding adversarial attacks

based on targeted object detection model designs, and then for each model design, we select the most effective attack design published so far to perform system-level effect measurement. Due to the availability of source code, we reproduce multiple STOP sign-hiding adversarial attack works. Then, we design a simulation-centric evaluation platform to perform the measurement study. More details will be introduced in §3. Our results show that all the representative existing works, can not cause any STOP sign traffic rule violation against a representative closed-loop control AD system in the common speed range for STOP sign-controlled roads in the real world even if the most effective attack can achieve more than 80% average attack success rate in general on the AI component alone.

We further explore the root causes and find that all the existing works have design limitation to achieve effective system-level effects due to lack of consideration of system model in AD context. We propose two design limitation hypotheses: 1) the unpractical STOP sign size distribution in pixel sampling, and 2) missing particular consideration in system-critical attack range, which will be detailed in §4. With that, we propose system model-driven attack design, which can be an addon of the existing attack methodologies to improve system-level effects by overcoming the two limitations.

We evaluate our attack improvement in the platform we designed and show that the system-level effect can be significant improved, i.e., the system violation rate can be increased around 70%. Ablation studies are also included in the evaluation, which shows the improvement on both component- and system-level for the setting with anyone of the hypothesis mentioned above and obtains the best results after applying both two hypotheses. Our code will be released after the double-blind review.

Note that we do not intend to claim to be the first to point out, analyze, measure, or optimize the gap between AI component errors and their system-level effect in general; there exists a large body of prior works in various other problem contexts (e.g., camera surveillance, video analytics, and control) across academia and industry that have studied the characterization, modeling, and/or optimization of end-to-end system performance with regard to AI/vision component errors (Jain & Binford, 1991; Ramesh et al., 1997; Thacker et al., 2008; Haralick, 1992; Ji & Haralick, 1999; Zhang & Zhu, 2018; Phillips et al., 2021; Greiffenhagen et al., 2000; 2001a;b; Philion et al., 2020; Caesar et al., 2020; Topan et al., 2022; Gog et al., 2021). Nevertheless, to the best of our knowledge, none of them (1) quantified such gaps in the context of adversarial attacks on autonomous systems, especially those in real-world system setups; and (2) identified novel designs that can systematically address/fill such gaps on autonomous systems, which we believe are our novel and unique contributions.

**Contributions**. To sum up, this paper makes the following contributions:

- We are the first to perform a comprehensive measurement study on the system-level effect of the representative prior works with the entire AD system pipeline with closed-loop control on our designed simulation-centric evaluation platform. Our results indicate that all the representative existing works, cannot cause any STOP sign traffic rule violation in common speed range for STOP sign-controlled roads in real world.

- We point out two design limitations of the prior works to hinder them in better achieving the system-level effects and propose system model-driven attack designs to overcome these.

- We further evaluate the validity of the two design limitations proposed in this paper and show that with our novel designs, the system-level effect can be significantly improved, i.e., the system violation rate can be increased around 70%

## 2 BACKGROUND AND SYSTEM MODEL DEFINITION

**Camera-based AD perception.** Today, camera-based AD perception generally leverages DNN-based object detection to recognize road objects of various categories (e.g., traffic signs, vehicles, pedestrians, and cyclists) in consecutive image frames (Carranza-García et al., 2020). State-of-the-art DNN-based object detectors can be generally classified into two categories: one-stage object detector, and two-stage object detector. The former, such as YOLO v2 (Redmon & Farhadi, 2017), YOLO v3 (Redmon & Farhadi, 2018), and YOLO v5 (Jocher, 2022), usually has higher detection speed, while the latter, such as Faster R-CNN (Ren et al., 2015), usually has higher detection accuracy. Since one-stage object detector processes bounding box (BBox) regression and object classification concurrently without a region proposal stage, it is generally much faster than two-stage ones and thus can better meet the real-time requirement in AD context (Carranza-García et al., 2020). In this paper, we focus on the security aspects of camera-based AD perception, and perform the corresponding experiments on both object detector categories.

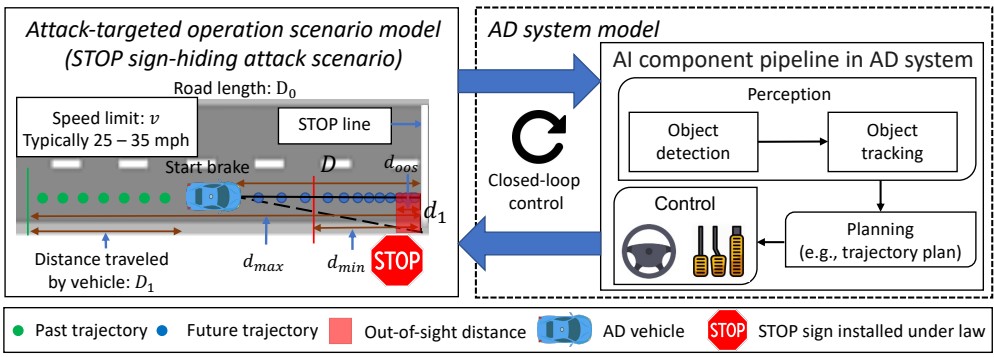

Figure 1: Illustration of the system model for STOP sign-hiding adversarial attacks in AD context. The minimum brake distance and out-of-sight distance are from the vehicle plant model.

**Physical object-hiding adversarial attacks in AD context.** Recent works find that DNN models are generally vulnerable to adversarial example, or *adversarial attacks* (Goodfellow et al., 2015; Papernot et al., 2016; Carlini & Wagner, 2017; Madry et al., 2017). Due to the direct reliance of camera-based AD perception on DNN object detectors, various prior works have explored the feasibility of adversarial attacks in such AD context (Jia et al., 2022; Zhao et al., 2019; Xu et al., 2020; Zolfi et al., 2021; Wang et al., 2021; Huang et al., 2020). Among them, *physical object-hiding adversarial attacks*, which typically use physical-world attack vectors such as malicious stickers/patches to cause the disappearance of important road objects (e.g., vehicles, pedestrians, and traffic signs) in the object detection results (Jia et al., 2022; Eykholt et al., 2018; Zhao et al., 2019; Xu et al., 2020; Chen et al., 2018; Wu et al., 2020), are especially severe due to their direct impacts on driving correctness and road safety. However, as detailed in later sections, we find that so far the considerations/integrations of the corresponding *system models* (detailed below) in the prior works in this research space are *far from enough* in both their attack designs and evaluation, which is found to substantially jeopardize the meaningfulness of their designs from the end-to-end AD driving perspective (§3). In this paper, we perform the first study to fill this critical research gap.

**Systems model for AD AI adversarial attacks.** To understand the end-to-end system-level impacts of an adversarial attack against a targeted AI component in an AD system (e.g., whether it can indeed effectively cause undesired AD system-level property violations such as collisions and traffic rule violations), we need to systematically consider and integrate the overall system semantics and context that enclose such AI component into the security analysis (Dreossi et al., 2019; Seshia et al., 2022). In this paper, we call a systematic abstraction of such system semantics and context the *system model* of such AD AI adversarial attacks. Specifically, in the AD context we identify 3 essential sub-components in such system model: (1) *the AD system model*, i.e., the full-stack AD system pipeline that encloses the attack-targeted AI components and closed-loop control; (2) *the vehicle plant model*, which defines the physical properties of the underlying vehicle system under control, e.g., maximum/minimum acceleration, deceleration, and steering rates, sensor mounting positions, etc.; and (3) *the attack-targeted operation scenario model*, which defines the physical driving environment setup, driving norms (e.g., traffic rules), and the system-level attack goal (e.g., vehicle collision, traffic rule violation, etc.) targeted by the AD AI adversarial attack.

**System model for STOP sign-hiding adversarial attack.** Fig. 1 illustrates the aforementioned system model defined for the STOP sign-hiding adversarial attack, which is so far the most extensively-studied physical object-hiding adversarial attack in AD context (Shen et al., 2022), and thus will be the main focus of our study in later sections due to such high representativeness in this research space. As shown, the AD system model for object detection, the targeted AI component in STOP sign-hiding adversarial attack, mainly includes its downstream tasks object tracking, planning, and control, and closed-loop control. The vehicle plant model mainly includes the physical properties related to longitudinal control, e.g., the minimum brake distance ($d_{min}$), and the distance to the stop line where the STOP sign is out of sight in the camera image $d_{oos}$ (depending on the hood length and the camera mounting position). The operation scenario model includes the speed limit, lane width of common STOP sign-controlled local roads, the relative positioning and facing of the STOP sign to the ego lane, the driving norm that the vehicle typically drives at constant speed before it starts to see the STOP sign ($d_{max}$), and the system-level attack goal that triggers the STOP sign violation (i.e., exceeding the stop line). We will use this system model in our studies in the following sections.

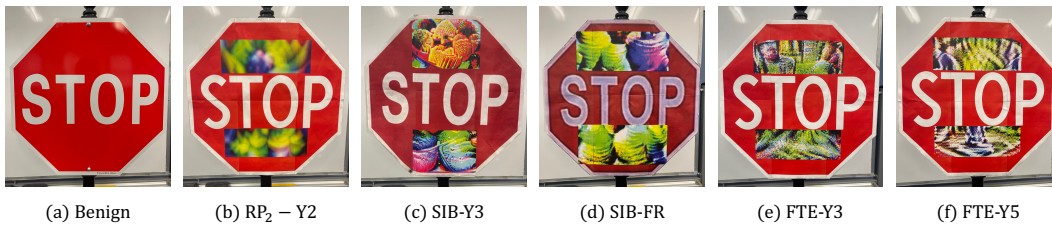

| (a) Benign | (b) $RP_2 - Y2$ | (c) SIB-Y3 | (d) SIB-FR | (e) FTE-Y3 | (f) FTE-Y5 |

Figure 2: STOP signs attack reproduction (in Table 1) visualisation used for measurement study, which are printed on ledger-size papers.

## 3 SYSTEM-LEVEL EFFECT MEASUREMENT OF PRIOR WORKS

**Scientific gap in existing works: Lack of system-level evaluation.** Despite a plethora of published attack works on physical object-hiding adversarial attacks in AD context (§2), we find that actually *all of them* only evaluate their attack effect *at the targeted AI component level* (i.e., judged by per-frame object misdetection rates (Eykholt et al., 2018; Xu et al., 2020; Zhao et al., 2019; Jia et al., 2022)), without any evaluation *at the system level*, i.e., with the corresponding system models for such attacks as described in §2. However, in the Cyber-Physical System (CPS) area, it is widely recognized that in AD system, AI component-level errors do not necessarily lead to system-level effects (e.g., vehicle collisions) (Dreossi et al., 2019; Seshia et al., 2022; Jia et al., 2020).

Thus, without system-level evaluation, it can be highly difficult, if not impossible, to scientifically know whether a proposed attack is actually meaningful from the end-to-end AD driving perspective. We view this as a critical scientific gap currently in this research space, and to address this, the necessary first step is to perform a measurement study on the existing works about their system-level effects. As the first study along this line, we choose to focus on *STOP sign-hiding adversarial attack* as our measurement target considering its high representativeness in this research space and also its direct impacts on driving correctness and road safety (§2).

### 3.1 ATTACK FORMULATION AND SELECTION OF PRIOR STOP SIGN ATTACK WORKS

**Attack formulation.** We assume that the attacker can arbitrarily manipulate pixels within restricted regions, which is known as well-defined adversarial patch attack (Brown et al., 2017; Zhao et al., 2019; Eykholt et al., 2018) in the prior works. Such a patch attack is easy to deploy in the real-world and very stealthy. We consider the patch stays on the STOP sign shown in Fig. 2

**Selection of prior STOP sign attack works.** There are various prior works on physical STOP sign-hiding adversarial attacks (Lu et al., 2017b; Jia et al., 2022; Eykholt et al., 2018; Zhao et al., 2019; Xue et al., 2021; Lu et al., 2017a; Chen et al., 2018). So far, *all* of them focus on studying the security of the AI component alone rather than with the entire AD system pipeline with closed-loop control. To perform our system-level effect measurement, we select the most effective ones at component level (i.e., AI models) as representative prior work examples. Four model designs have been covered in these prior studies, which generally belong to 2 types (§2): one-stage (e.g., YOLO family) and two-stage (Faster RCNN) object detection. For each model, we select the most effective attack design published so far. The attack selection and the its model designs are in Table 1.

### 3.2 MEASUREMENT METHODOLOGY AND SETUP

To measure system-level effects, we adopt a simulation-centric evaluation methodology (details in Appendix A), which has been widely adopted both in academia (Wan et al., 2022; Sato et al., 2021) and in industry (Way, 2021; Sca, 2021) today due to the inherent limitations of real-road AD testing in cost, safety, efficiency, and corner-case coverage. Thus, we believe that using simulation for system-level evaluation is on par with the best and validated practices in both academia and industry.

**Simulation fidelity.** In this paper, we use SVL, an production-grade high-fidelity AD simulator designed specifically for evaluating production-level AD systems (Rong et al., 2020). It leverages Unity's built-in physics engine to accurately simulate the vehicle dynamics and tire-road interaction, and provide photo-realistic simulation of the driving environment that closely matches the real world (Rong et al., 2020). As repeatedly demonstrated in various prior works, the end-to-end AD system-level evaluation results in SVL, especially those for adversarial AI attacks, can highly cor-

Table 1: Selection of prior works to be evaluated in our system-level effect measurement study. Specifically, for each of the 4 model types targeted by prior works, we select the most effective attack design published so far. More details are in §3.1

| Model | YOLO v5 (Y5) | YOLO v3 (Y3) | YOLO v2 (Y2) | Faster RCNN (FR) |
|---|---|---|---|---|
| Attack | FTE (Jia et al., 2022) | SIB (Zhao et al., 2019) | $RP_2$ (Eykholt et al., 2018) | SIB (Zhao et al., 2019) |

Table 2: B: benign; ASR: attack success rate. System-level evaluation in the simulation-based testing (§3.2) and component-level overall ASR for model Y2, Y3, Y5, and FR in benign, $RP_2$-, SIB-, and FTE-attacked scenarios in common speed ranges for STOP sign-controlled roads. The results are the STOP sign violation rate. Each speed contains 10 runs with different initialization of the AD vehicle initial position. For system-level violation rate, we only include the results in which the benign cases perform 0% violation rate. We calculate the component ASR for attacked cases.

| Evaluation level | Speed (mph) | Y2 | | Y3 | | | Y5 | | FR | |
|---|---|---|---|---|---|---|---|---|---|---|
| | | B | $RP_2$ | B | SIB | FTE | B | FTE | B | SIB |
| System (violation rate) | 25, 30, 35 | 0% | 0% | 0% | 0% | 0% | 0% | 0% | 0% | 0% |
| Component (ASR) | Overall | - | 80.1% | - | 79.7% | 59.9% | - | 46.1% | - | 5.8% |

relate with the same setup tested in the physical world (Wan et al., 2022; Sato et al., 2021). In our paper, to even further ensure the fidelity of our evaluation results, we further improved the fidelity of the rendering process by modeling the perception results in the real world with a practical setup (Appendix A). Similarly, such high simulation fidelity has also been justified multiplied times for the control process. For instance, a research team at UC Berkeley has tested several representative scenarios generated in SVL (Fremont et al., 2019) in a physical vehicle testing track, and concluded that SVL is "effective at synthesizing test cases that transfer well to the track" (Fremont et al., 2020).

### 3.3 MEASUREMENT RESULTS

We first evaluate our reproduction correctness. More details about the results are in Appendix B

With perception results modeling, we inject the detection rates measured under benign and attacked scenarios to the AD system (§3.2) to evaluate their targeted system-level attack effect, i.e., STOP sign violation rate. We define the STOP sign violation rate as $\frac{N_{\text{violation}}}{N_{\text{total}}}$, in which $N_{\text{violation}}$ means the number of runs where the AD vehicle exceeds the STOP line and $N_{\text{total}}$ is the number of total runs. Table 2 shows the results where each speed has 10 runs with random initialization of the AD vehicle position. To our surprise, *none* of the existing representative attacks were able to trigger STOP sign violations in *any* of the common speeds for STOP sign-controlled roads when the benign performs well, even if most of the attacks are effective in the component (i.e., with over 54% average attack success rate across the 5 attacks). After inspecting the details, we find that this is because the STOP sign is always tracked at the object tracking step before reaching the minimum brake distance of the AD vehicle. Taking $RP_2$-Y2 as an example, the brake distance for 25 mph is around 10 m, and in the benign scenario, this is in the range (5-10m) where the detection rate in benign scenarios is 100% as shown in Table 3. The $RP_2$ attack can only reduce the detection rate to 90% (i.e., only 10% attack success rate) as shown in Table 3. Thus, it is not enough to make the tracking vanished before the minimum braking distance. Thus, the AD vehicle can always fully stop before the STOP sign.

### 4 DESIGN LIMITATION HYPOTHESES AND IMPROVEMENT PROPOSAL

After finding out that the existing works cannot lead to any system-level violation We find out that the prior works are not fully leveraging the information from the system model and thus, fail to perform effective system-level effects. We further investigate the system model in AD context and propose two design limitation hypotheses which leads to low effectiveness in the system-level evaluation.

#### 4.1 DESIGN LIMITATION HYPOTHESIS I

In the prior works (Zhao et al., 2019; Jia et al., 2022), to make the attack robust to different object sizes, Expectation over Transformation (EoT) (Athalye et al., 2018) is usually used to uniformly sample the object size in a certain range (Chen et al., 2018; Jia et al., 2022; Athalye et al., 2018).

Table 3: Detection rates of YOLO v2 (Y2), YOLO v3 (Y3), YOLO v5 (Y5), and Faster-RCNN (FR) in benign, RP$_2$-, SIB-, and FTE-attacked scenarios from our physical-world experiments. Each detection rate below is calculated with 400 video frames.

| Object Detector | | Distance range (m) | | | | | | | | |
|---|---|---|---|---|---|---|---|---|---|---|
| | | 4 - 5 | 5 - 10 | 10 - 15 | 15 - 20 | 20 - 25 | 25 - 30 | 30 - 35 | 35 - 40 | 40 - 45 |
| Y2 | Benign | 100% | 100% | 71.3% | 31.3% | 0% | 0% | 0% | 0% | 0% |
| | RP$_2$ | 58.2% | 90.0% | 76.2% | 34.6% | 0.1% | 0% | 0% | 0% | 0% |
| Y3 | Benign | 100% | 100% | 100% | 100% | 80.1% | 11.8% | 6.7% | 1.0% | 0% |
| | SIB | 93.7% | 100% | 100% | 90.4% | 38.2% | 0% | 0% | 0% | 0% |
| | FTE | 89.9% | 100% | 100% | 87.3% | 42.9% | 0.6% | 0% | 0% | 0% |
| Y5 | Benign | 100% | 100% | 100% | 100% | 98.7% | 89.4% | 52.3% | 25.3% | 0% |
| | FTE | 91.2% | 100% | 100% | 99.7% | 88.2% | 48.4% | 3.9% | 0% | 0% |
| FR | Benign | 100% | 100% | 100% | 100% | 100% | 100% | 100% | 100% | 100% |
| | SIB | 100% | 100% | 100% | 100% | 100% | 100% | 100% | 100% | 53.2% |

(a) Distribution of exiting work    (b) Distribution from simulation    (c) Distribution from theoretical analysis

Figure 3: Different STOP sign size distribution. The distribution from the (a) existing work (Jia et al., 2022), (b) our experimental analysis on H1, and (c) our theoretical analysis on H1.

However, considering the system model in AD context, we find that *such uniform sampling distribution is actually inaccurate*, which leads to our first design limitation hypothesis:

*Design Limitation Hypothesis I (H1): the STOP sign size in pixel sampled distribution is not uniform in the attack's system model (§2) when the vehicle is moving towards the STOP sign.*

**Experimental analysis of H1.** With the same setup in §3, we can simulate the real driving scenario. During the driving, the STOP sign size in pixels and the distance between the vehicle and the STOP sign can be directly obtained from the High-Definition (HD) map in the AD system (§3.2). With that, we can easily plot the frequency distribution histogram over different STOP sign sizes in pixel. Such distribution is shown in the Fig. 3 (b) after normalization, where the AD vehicle in the simulation runs for 30 rounds at 25 mph. The distribution is totally different from the uniform distribution used in the prior works (Chen et al., 2018; Jia et al., 2022). To compare, we also sample the STOP sign size in the most recent prior work (Jia et al., 2022), in which they design an algorithm to determine the STOP sign size in a uniform way. We run that algorithm $3 \times 10^4$ times and collect the STOP sign size. The difference between Fig. 3 (a) and (b) indicates that our H1 is held experimentally.

**Theoretical analysis of H1.** Assuming the AD vehicle performs uniform motion toward a STOP sign, we can leverage the camera pin-hole model as shown in Fig. 4 to perform the theoretical analysis. Based on the Fig. 4, we can easily get the relationship of real object size ($L$), real distance($D$), focal length($f$), and object size in pixel($s$) with similar triangles: $\frac{L}{D} = \frac{s}{f}$. With the system model shown in Fig. 1, we assume that the initial vehicle to STOP sign distance is the road length $D_0$ and the current vehicle to STOP sign distance is $D$. Due to uniform motion, the vehicle travelled distance can be formulated as $D_1 = v * t$, where $v$ is the vehicle speed (usually it is the speed limit) and $t$ is the time. To know the relationship between the STOP sign size and the sampled frequency (i.e., the frame number), we formulated the time $t$ as $t = \frac{F}{\eta}$, where the $F$ is the number of frames and the $\eta$ is the image capturing frequency from the camera with unit frame/s. Due to $D_1 + D = D_0$ and the similar triangles in the pin-hole model, we can easily obtain the following equation:

$$D_0 = D + v * \frac{F}{\eta} = \frac{L * f}{s} + v * \frac{F}{\eta} \rightarrow F = (D_0 - \frac{L * f}{s}) * \frac{\eta}{v} \tag{1}$$

Thus, from Eq. (1), we can easily get cumulative distribution function (CDF) of $s$ since the $F$ here is the accumulated frames. To get the probability density function (PDF), we calculate its derivative:

$$F' = \frac{dF}{ds} = \frac{\eta * L * f}{v * s^2} \tag{2}$$

From Eq. (2), the distribution over the size of the STOP sign in the pixel is not uniform, which proves the H1. We also plot the curve with Eq. (2) as shown in Fig. 3 (c), where we use the $\eta = 20$, $L = 1.5$, $v = 25mph$, and $f = 25mm$ (the parameters here are commonly used in AD system such as Baidu Apollo). The distribution curve is very similar to the distribution we find in the experimental analysis shown in Fig. 3 (b), which shows that the H1 is held.

## 4.2 DESIGN LIMITATION HYPOTHESIS II

In the EoT process of prior works, uniformly sampling the STOP sign size in a range is generally used. However, without the system model ( §2), it is difficult to precisely obtain the STOP sign size range, and thus, in the prior works, they just treat these as hyper-parameters but without any reasoning for selection (Chen et al., 2018; Jia et al., 2022). However, in prac-

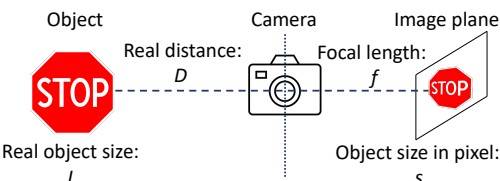

Figure 4: Theoretical analysis on H1, i.e., the camera pin-hole model.

tice, not every range is equivalent to achieving the system-level effects. For instance, within $d_{min}$ in Fig 1, even if the AD vehicle applies the maximum deceleration, it still cannot fully stop before the STOP line (i.e., exceed the minimum brake distance), then the attack effectiveness does not matter in such a range, which indicates that such a range is not important to achieve system-level effects. However, none of the prior works consider such a system-critical range in their designs due to a lack of consideration of the system model. This thus comes to our second design limitation hypothesis:

*Design Limitation Hypothesis II (H2): Prior works generate the attack without considering the system-critical range systematically (can be obtained from the system model in §2), which leads to low effectiveness in the system-level effect.*

Note that here blindly using a large range of STOP sign size to attack, it will provide lower attack effectiveness compared to the small certain range of STOP sign size. For instance, when the STOP sign size is small (common cases when the AD vehicle is far away from the STOP sign), it is very difficult to make the attack converge (Jia et al., 2022), which indicates that it is harder to perform the attack. Thus, in generally, if we can find a more optimal range rather than a super large range to generate the attack, it will benefit the attack effectiveness. We also have experimental analysis on this claim and more details can be found in the Appendix E. To validate H2, we will design the experiments and ablation study with our improvement proposal in §4.3 to further evaluate it in §5.

## 4.3 PROPOSAL: SYSTEM MODEL-DRIVEN ATTACK DESIGN

In order to overcome the two design limitations mentioned above, we propose the following improvement based on the system model in the AD context. Here, our improvement proposal is to design an addon of existing attacks, which is orthogonal to the original attack designs and can better overcome the two limitations to achieve high system-level effectiveness.

To overcome H1, we apply a new transformation distribution. As shown in Eq. (2), the relationship between the number of frames and the STOP sign size is $F' \propto \frac{1}{s^2}$, where $F'$ is the number of frames and $s$ is the STOP sign size in pixel. Thus, we define $\mathcal{S} = \{s_1, s_2, ..., s_N\}$ as a discrete distribution, where $s_i$ is the a STOP sign size in pixels. Based on Eq. (2), the probability of $s_i$ is $p(s_i) = \frac{1}{s_i^2} / \sum_{k=1}^{N} \frac{1}{s_k^2}$. With that, we can apply this new distribution with a new objective:

$$\arg\min_{p_a} \quad \mathbb{E}_{s \sim \mathcal{S}}[\mathcal{L}(M(p_a, O, s), *)] \tag{3}$$

The $\mathcal{L}$ is the loss function used in the prior attacks, $p_a$ is the adversarial patch, $O$ is the STOP sign, and function $M(p_a, O, s)$ indicates applying the adversarial patch $p_a$ to the STOP sign $O$ and resizing the STOP sign size in pixel to $s$, $*$ means other inputs for loss function in the prior works such as the bounding box information, detection threshold, and $\mathcal{S}$ is the distribution mentioned above. With Eq. (3), we can simulate the distribution obtained from the real world to balance the attack success rate between the far distance and near distance.

However, one missing part for the Eq. (3) is the STOP sign size but such important information can be obtained from the system model (Fig. 1) by overcoming H2 (details in Appendix D). With the system-critical range, the next step is to transfer the system-critical range in the physical world to the pixel range in the image. We use a camera-based rendering technique and leverage nuScenes dataset (Caesar et al., 2020) to achieve our goal since it directly provides some APIs to render an

Table 4: Attack success rates of $RP_2$-Y2, FTE-Y3, and FTE-Y5 for system model-driven attack design evaluation from our physical-world experiments. Each detection rate below is calculated with 600 video frames.

| | | Distance (m) | | | | | | | | |
|---|---|---|---|---|---|---|---|---|---|---|
| | | 4 - 5 | 5 - 10 | 10 - 15 | 15 - 20 | 20 - 25 | 25 - 30 | 30 - 35 | 35 - 40 | 40 - 45 |
| | H1 | 4.4% | 13.7% | 51.2% | 99.3% | 100% | 100% | 100% | 100% | 100% |
| $RP_2$-Y2 | H2 | 5.6% | 44.9% | 57.8% | 98.7% | 100% | 100% | 100% | 100% | 100% |
| | H1 + H2 | 36.1% | 65.8% | 88.0% | 100% | 100% | 100% | 100% | 100% | 100% |
| | H1 | 0% | 0% | 0% | 14.0% | 72.2% | 95.9% | 100% | 100% | 100% |
| FTE-Y3 | H2 | 0% | 0% | 0% | 13.4% | 81.4% | 94.4% | 97.2% | 100% | 100% |
| | H1 + H2 | 5.3% | 0% | 34.7% | 94.0% | 99.4% | 100% | 100% | 100% | 100% |
| | H1 | 0.3% | 0% | 0% | 1.3% | 32.7% | 81.9% | 94.1% | 99.0% | 100% |
| FTE-Y5 | H2 | 1.5% | 0% | 0.1% | 1.7% | 13.9% | 69.3% | 99.0% | 100% | 100% |
| | H1 + H2 | 16.5% | 0% | 4.3% | 47.2% | 93.4% | 99.7% | 100% | 100% | 100% |
| | H1 + H2 (TV) | 43.6% | 51.7% | 42.1% | 26.3% | 23.8% | 66.1% | 97.7% | 99.7% | 100% |

object into an image. Specially, we render the four corners of the STOP sign with physical-world properties (e.g., real-world size) and obtain the STOP sign size in pixel by measuring the distance between these four points in the image, i.e., the height and width. With Eq. (3), we can embed the system-model property into the attack generation to improve the system-level effects.

## 5 HYPOTHESIS VALIDATION

We adopt a same evaluation methodology and setup used in §3.2 but just replace the STOP sign patch with the newly generated ones. In this section, we only select some of the attacks on one-stage object detectors, i.e., YOLO-based object detectors including Y2, Y3, and Y5, since one-stage object detectors have better real-time performance than two-stage ones and they are used in the Autoware.AI (Kato et al., 2018), an industry-grade full-stack AD system. With that, we select $RP_2$ and FTE as the corresponding attacks due to their representativeness discussed in §3.1. The detailed combination for object detectors and attacks are as follows: $RP_2$-Y2, FTE-Y3, and FTE-Y5.

### 5.1 SYSTEM MODEL-DRIVEN ATTACK DESIGN EVALUATION AND ABALTION STUDY

**Attack generation.** We adopt the methodology in §4.3, of which details are in Appendix F.

**Results analysis.** The STOP sign attack images are as shown in Fig. 6 from Appendix G. As shown in Table 5, with our system model-driven attack designs, the system violation rate can increase by around 70% on average. Note that in Table 5, we only include the results where the benign cases can perform 0% system-level violation rate. With H1 + H2, the overall component attack success rate can increase around 12% on average due to the practical setup. Especially, in the system critical range, the attack success rate can increase by 38%, which can significantly improve the system-level effects. Taking FTE-Y5 at 35 mph as an example, the brake distance of 35 mph is around 20 m and the attack success rate from 20 - 35 m shown in Table 4 is around 98%, which indicates that it has a very high chance to make the STOP sign not tracked before the brake distance. This leads to the 100% system violation rate shown in Table 5. For FTE-Y5 at 25 mph, due to the low effectiveness (i.e., around 4%) from 10 m to 15 m, it is very difficult to achieve any system-level effects because of the tracking, which leads to a 0% system violation rate. Based on results in Table 4, in general, the attack success rate in a near distance between the STOP sign and the vehicle is lower (i.e., more difficult to attack), which aligns well with the prior work results (Zhao et al., 2019). This leaves a space for future works that could improve the AI component attack success rate in near distance. In this paper, we provide small improvement for FTE-Y5 at 25 mph shown in §H.

The results for the ablation study are also summarized in Table 5. Although in the majority of cases, H1 only cannot significantly improve the system-level effects (20% on average), the component attack success rate in the system-critical range is improved, which further demonstrates the usefulness of the H1. Compared to H1, H2 has better system-level effects (around 30% on average) and attack success rate in the system critical range, but it still cannot achieve significant improvement on the system-level effects. Combining H1 and H2 can further benefit the results for system-level effects (around 70% on average), which shows the necessaries of both H1 and H2.

Table 5: None: without H1 or H2; H1: with H1 only; H2: with H2 only; H1 + H2: with H1 and H2; SCR: System-critical range (§4.2). System-level violation rate (with P-Value) and component-level ASR evaluation for system model-driven attack design evaluation and its baseline (i.e., prior works and ablation studies). Each speed contains 10 runs with different initialization of the AD vehicle initial position in the system-level evaluation. The number of frames is over 600 for the component-level evaluation (Table 4). Generally, with our H1 and H2, the system-level violation rate and component ASR (especially in SCR) is improved compared to prior works and its ablation studies. "Nan" in P-Value calculation is caused by the means are the same for the two samples.

| Evaluation level | Speed (mph) | RP$_2$ | | | | FTE-Y3 | | | | FTE-Y5 | | | |
|---|---|---|---|---|---|---|---|---|---|---|---|---|---|
| | | None | H1 | H2 | H1 + H2 | None | H1 | H2 | H1 + H2 | None | H1 | H2 | H1 + H2 |
| System (violation rate) | 25 | 0% | 90% | 100% | 100% | 0% | 0% | 0% | 40% | 0% | 0% | 0% | 0% |
| | 30 | - | - | - | - | 0% | 0% | 30% | 100% | 0% | 0% | 0% | 80% |
| | 35 | - | - | - | - | - | - | - | - | 0% | 30% | 40% | 100% |
| P-Value | | - | $4.4 \times 10^{-8}$ | 0.00 | 0.00 | - | Nan | 0.08 | $7.2 \times 10^{-8}$ | - | 0.08 | 0.04 | $1.4 \times 10^{-8}$ |
| Component (ASR) | Overall | 80.1% | 74.2% | 78.6% | 87.8% | 59.9% | 53.6% | 54.0% | 70.4% | 46.1% | 45.1% | 49.4% | 62.3% |
| | SCR | 33.1% | 54.7% | 67.1% | 84.6% | 33.8% | 36.4% | 37.8% | 65.6% | 26.6% | 34.2% | 40.8% | 57.4% |

**Statistical significance.** To quantify the statistical significance, we calculate the P-Value in Table 5 for all trials compared with the None column. The P-Value is generally at the statistically significant level (e.g., generally $< 0.05$ or at a similar magnitude, especially for the most important H1+H2).

We also improve the system-level results for low speed one by leveraging the total variation (TV) loss as prior works (Eykholt et al., 2018; Cao et al., 2021), which is discussed in the Appendix H.

## 6 GENERAL RESEARCH TAKEAWAYS

We summarize various general research takeaways from this research effort beyond the specific application scenario that we evaluate on: (1) From the measurement point of view, *in AD systems, the gap between the component-level and system-level attack effects can be much larger than what prior works thought.* For instance, SIB paper (Zhao et al., 2019) claims that "our AEs could potentially cause serious problems for autonomous driving cars." However, in our paper, we evaluated the SIB in our practical AD setup and found that it can actually only achieve 0% system-level violation rate. Such a huge gap in this domain clearly suggests the need for system-level evaluation like what we performed in §3. (2) From the new attack methodology design point of view, *our two validated hypotheses are both generalizable to improve the system-level attack effect in other AD attack settings.* For instance, if considering a pedestrian-hiding attack (instead of STOP sign), due to the motion of the vehicle, the different distribution and the system-critical range both still exist, and thus systematically considering them in the attack generation can help to improve the system-level effect. (3) From the general solution direction point of view, *our system model-driven designs also provide general takeaways from secure/robust model training perspective.* To train the AD AI model to avoid undesirable behaviors at the system level, it is highly desired to first have such concrete attack examples that can be meaningful/effective at the system level. In our paper, we improve the AD system-level effects of the adversarial AI attack generation process with novel systematic integration of system models. The system models may inspire the integration of the system model into further model training process to generally improve model security, robustness, or even accuracy in any AI-enabled systems instead of just for AD.

## 7 CONCLUSION

In this paper, we propose an interesting and important research question: can previous works actually achieve system-level effects (e.g., vehicle collisions, traffic rule violation) under real-world AD settings with closed-loop control? To answer such a question, we perform the first measurement study on whether and how effective the existing designs can lead to system-level effects, where we take the STOP sign-hiding attack as our target. Our evaluation results show that all representative prior works cannot achieve any system-level effect in a classical closed-loop AD setup in road speeds controlled by common STOP signs due to lack of consideration of the system model. Experimental results demonstrate that with the system model-driven designs, the system-level effects can be further improved, i.e., the violation rate can increase around 70%. We hope that the concept of the system model could guide further security analysis to achieve better system-level effects.

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

# A   MEASUREMENT METHODOLOGY AND SETUP DETAILS

**Evaluated AD system pipeline.** In this paper, we design a simulation-centric testing, including the simulation-based evaluation setup with SVL simulator, an production-grade Unity-based AD simulator (Rong et al., 2020), leveraged by prior works to perform evaluation in AD context (Cao et al., 2021; Wan et al., 2022; Zhang et al., 2022; Hallyburton et al., 2022). The comparison of SVL simulation results and real runs indicate SVL provides certain level fidelity (Fremont et al., 2020). In SVL, we perform the experiment with the San Francisco map in a sunny day at noon shown in Fig. 5 (b) from Appendix §C as the operation scenario in Fig. 1. The AD system pipeline that we use includes representative downstream tasks and setups after object detection, which includes (1) a tracking step using a general Kalman Filter based multi-object tracker (Luo et al., 2021), (2) a planning step using a lane-following planner from Baidu Apollo (Apollo, 2022), an industry-grade full-stack AD system, and (3) a control step using classic controllers such as PID for longitudinal control used in OpenPilot (OpenPilot, 2022), a production Level-2 AD system, and Stanley (Hoffmann et al., 2007) for lateral control.

**Attack reproduction.** All the STOP sign attacks that we want to investigate in Table 1 do not provide the source code. We tried to contact the authors of the attack in Table 1 for the source code, but they all cannot provide it. Thus, we try our best to reproduce some of the works and will release our reproduction in the future to benefit future researchers. Currently, we only have the reproduction for $RP_2$ and FTE. For SIB, the authors of that paper share the STOP sign images that they used in their physical-world experiments. Thus, we directly use the ones provided by them. We print the real-world high-resolution STOP signs on multiple ledger-size papers and concatenate them together to form a full-size real STOP sign. The STOP signs used for measurement study are shown in Fig. 2

**Perception results modeling.** To better address the perception fidelity in simulators, we model the perception results in the real world with a practical setup. We did not directly apply the setup from the existing works (Zhao et al., 2019; Jia et al., 2022) due to their unrealisticness. In the prior works, when they collected the video frames, they would directly move towards to the STOP sign and change the angles based on the rotation of the STOP sign. Such a setup is not practical since in the real world, the AD vehicle will not directly drive towards the STOP sign and the STOP sign should be located on the roadside as shown in Fig. 1. To improve such unrealistic setups, we follow the system model that we defined in §2 to put the STOP sign on the road side and control the movement of AD vehicle in the road center. We recorded several pieces of video along the $D$ using an iPhone 12 Pro Max starting from 45 m to 4 m (4 m is the $d_{oos}$ introduced in §2). We choose 45m since (1) it is a brake distance for above 50 mph, which exceeds the usual maximum speed of STOP sign areas, and (2) it is already much larger than the maximum distance evaluated in all the prior STOP sign-hiding attack works. We separate the whole range into 9 pieces, each spanning 5 m except the one near the STOP sign, which is 1 m from 4 m to 5 m. Then, we record a video in each region and feed the video into the object detectors in Table 1. Note that for each region, we collect more than 400 video frames, which is much larger than prior works (Chen et al., 2018; Eykholt et al., 2018; Jia et al., 2022). We perform these experiments on sunny days. The sample photo of collecting data for modeling perception results is shown in Fig. 5 (a) from Appendix §C. With the real world-measured STOP sign detection rate in each 5 m long range, we perform perception results injection at the output of the object detection task in our created AD system following such detection rate, i.e., first read the ground-truth STOP sign detection results from the simulator and then drop/keep the detection results based on detection rate. For instance, if the attack success rate is 60% for that distance range and the vehicle to STOP distance is within that range, for the object detection output, we will have 60% possibility to drop that STOP sign detection results, i.e., remove the STOP sign detection from the detection output.

**Speed selection.** The driving speed we select is from 25 to 35 mph, with a step size of 5 mph, which is the most common speed range for STOP sign-controlled roads in the real world. Specifically, 25 mph (Atlanta, 2020) is the common speed limit for the STOP sign-controlled road intersections, which is more likely to avoid a crash. On the other hand, 35 mph (California, 2022) is the most common speed limit for city streets (i.e., which STOP sign are designed for).

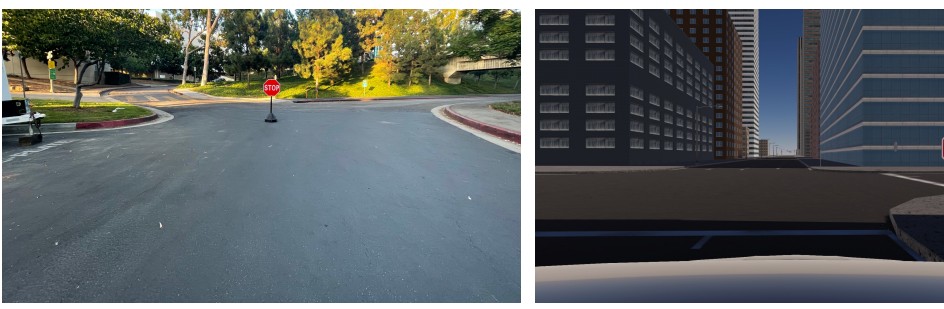

(a) Physical-world scene                    (b) Simulation scene

Figure 5: Scene for the experiment. (a) Real-world scene for collecting the data including real road and injected STOP sign (b) SVL simulation scene in the San Francisco map with STOP sign.

## B  STOP SIGN ATTACK REPRODUCTION RESULTS

For the STOP sign reproduction shown in Fig. 2, we follow the original paper's description with the new setup for perception results modeling and the results are as shown in Table 3. Note that we also try our reproduction with their original evaluation setups and find that the results are generally similar, which thus shows that our reproduction is correct. For instance, the original $RP_2$ paper (Eykholt et al., 2018) reports around 63.5% attack success rate from 0 - 10 feet, while if we follow the same setup (outdoor environment) as the original paper (less practical setup in AD context), we can achieve 61.0% attack success rate which is very close to the original results. Note that here the SIB attack with FR object detector looks weird since only from 40 - 45 m, the attack has around 47% attack success rate while in other range, it is always 0% attack success rate. Although we directly use the patch provided by the authors, the FR object detector could be different, where we use the MMDetection (Chen et al., 2019), an open source object detection toolbox based on PyTorch. Thus, leveraging the low transferability, the attack may not be effective compared to the results shown in original paper. However, this is already our best efforts to reproduce their results.

## C  SCENE FOR THE EXPERIMENTS

Fig. 5 shows the scene for our experiment in physical-world and simulation-based testing environment.

## D  METHODOLOGY FOR OBTAINING THE STOP SIGN SIZE

One missing part for the Eq. (3) is the STOP sign size and such important information can be obtained from the system model (Fig. 1) by overcoming H2. When it comes to H2, there must be a system-critical range to generate the attack with the system model. Several components in the AD system affect the range, such as object detection, object tracking, and control (e.g., brake). Such designs are all AD system specific and thus to obtain a general system-critical range, we use a reasonably large range to generate the attack. As shown in the attack's system model in Fig. 1, the $d_{min}$ of such a system-critical range definitely will be the minimum brake distance, since within the brake distance the detection results will not affect the system-level effects. When it comes to the $d_{max}$, many tasks in the AD system will be considered. For object detection, the maximum distance should be the longest distance for the benign object detector to detect the STOP sign, which is measured in Table 3. For tracking, we consider a very conservative tracking Jia et al. (2020) to perform the attack since the attacker in the real world may not directly get the tracking parameter in the targeted AD system and a conservative tracking provides a larger system-critical range, which can cover the system-critical range in general cases. In order to achieve system-level effects, the STOP sign should not be tracked when the vehicle reaches the brake point. Due to taking a very conservative tracking in the AD system (introduced in §3.2), such tracking distance (i.e., if within this distance, the STOP sign can never be detected, the tracker will delete the STOP sign) is usually larger than the distance where the object detector can detect the benign STOP sign. Thus, we can

Table 6: RP$_2$ attack success rate modeling in the simulation environment. The attacks are generated with different STOP sign size range in pixel. Small: the small range of the STOP sign is from 30 to 100. Large: the large range of the STOP sign is from 30 to 416, i.e., the largest range in which the benign STOP sign can be detected by the object detector, i.e., Y2.

| | | Distance (m) | | | | | | | Average |
| | | 4 - 5 | 5 - 10 | 10 - 15 | 15 - 20 | 20 - 25 | 25 - 30 | 30 - 35 | |
|---|---|---|---|---|---|---|---|---|---|
| Component ASR | Small | 6.7% | 37.1% | 68.3% | 81.1% | 100% | 100% | 100% | 70.5% |
| | Large | 98.6% | 6.1% | 0% | 1.0% | 58.5% | 99.1% | 100% | 51.9% |

Table 7: Detailed settings for attack parameters

| Parameter | Value | | |
| | RP$_2$ | FTE-Y3 | FTE-Y5 |
|---|---|---|---|
| Attack iteration | $3 \times 10^4$ | $6 \times 10^4$ | $2 \times 10^4$ |
| Initial learning rate | 0.5 | 0.03 | 0.05 |
| System-critical range (m) | (5, 20) | (5, 30) | (5, 35) |
| Batch size | 1 | 6 | 16 |
| $\lambda$ in (Eykholt et al., 2018) | 1.0 | - | - |
| (c, k) in (Jia et al., 2022) | - | (100, 10) | (100, 10) |
| GPU device | RTX 3090 | RTX 2080 Ti | RTX 3090 |

simplify this process and select an effective range as minimum brake distance ($d_{min}$) and the first distance ($d_{max}$) where the benign STOP sign can be detected with small detection rate.

## E  BASELINE EXPERIMENTS FOR H2

In this section, we perform the experiment on the attack with small and large range of the STOP sign size in the attack generation EoT part, and compare them.

We follow the similar evaluation setup as in §3 but use a pure simulation-based setup and measure the attack success rate across different ranges with RP$_2$ attack. The small range of the STOP sign is from 30 to 100, which is a critical range shown in §4, while the large range of the STOP sign is from 30 to 416, i.e., the largest range in which the benign STOP sign can be detected by the object detector, i.e., Y2. We generate the STOP sign attack and run them in the simulation.

Table 6 shows the results. The small range one has higher attack success rate compared to the large one, especially within the system-critical range shown in Table 7. The large range one seems to converge well in small distance (i.e., when the STOP sign is very near to the AD vehicle) but performs worse from 5 to 30 m. This indicates that it is very difficult to directly set large range to achieve better performance in the system-critical range.

## F  DETAILED ATTACK GENERATION

We adopt the methodology introduced in §4.3. The detailed attack generation parameters are shown in Table 7. Based on §4.3, we can easily calculate the system-critical range for the attack. For example, in RP$_2$, the brake distance for 25 mph is around 10 m, and considering that a safety buffer is usually applied in AD system (Apollo, 2022), thus, the $d_{min}$ for the system-critical range that we use is 5 m (considering a buffer distance). For the $d_{max}$, the STOP sign can be detected at around 20 m, which could be used directly to generate the attack.

In order to further validate the effectiveness of our two proposed design limitation hypotheses, we perform ablation studies. We generate the attack with the system model-driven attack design on H1 only, and H2 only, and compare them to the attack generated with/without both H1 and H2. Details of attacks without H1 and H2 are in §3. For the details of H1 only, we use our new distribution (in 4) with EoT introduced in §4.1 and for H2 only, we use the range from Table 7.

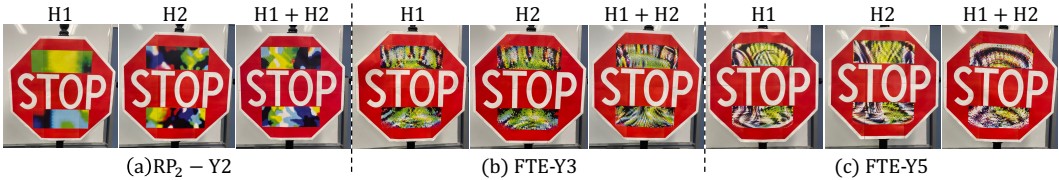

Figure 6: STOP signs attack generation with system model-driven design and their ablation studies. H1: attack only with H1; H2: attack only with H2; and H1 + H2: attack with both H1 and H2.

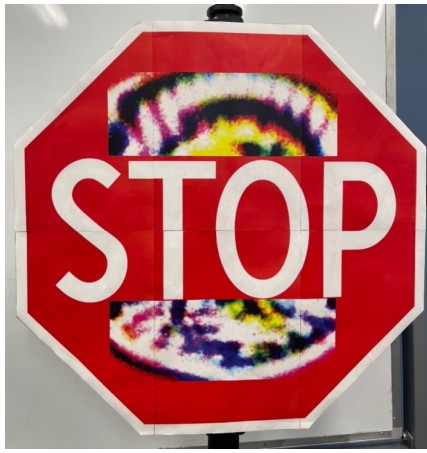

Figure 7: FTE-Y5 H1 + H2 with TV loss STOP signs attack generation, which is printed on ledger-size papers.

## G    STOP SIGN VISUALIZATION FOR §5

The STOP sign visualization for §5 is in Fig. 6

## H    IMPROVEMENT AT THE LOW SPEED

**Methodology and setup.** the FTE-Y5 attack at 25 mph speed has a 0% system-level violation rate (Table 5) due to the ineffective attack (Table 4) from 5 - 20 m. We apply the total variation (TV) loss as prior works (Eykholt et al., 2018; Cao et al., 2021) to improve the smoothness and thus, benefit the attack effectiveness in a larger range. All the setups are the same as the ones in §5.1.

**Results analysis.** The perception modeling results are shown in Table 4 and the STOP sign with the new patch is shown in Fig. 7. The attack success rate from 4 - 15 m is improved around 7 times though the attack success rate from 15 - 40 m becomes worse. We try this modeling results in the AD system (§3.2) which is the same setup as the one in §5.1 and find that at 25 mph, the system violation rate is 10% for 10 runs. Based on the results in Table 4, it is not trivial to balance the attack success rate between far distance and near distance, which could be a future direction.

