# OpenReview forum: "On the System-Level Effectiveness of Physical Object-Hiding Adversarial Attack in Autonomous Driving"
_ICLR.cc/2023/Conference — Submitted to ICLR 2023_

### Official Review · Reviewer_A81f · 2022-10-25

**Confidence:** 3
**Correctness:** 3
**Technical Novelty And Significance:** 2
**Empirical Novelty And Significance:** 2
**Recommendation:** 5

**Clarity, Quality, Novelty And Reproducibility:**

On the novelty side, the impact of perception errors in the context of AD system has been studied by several recent works (e.g., Learning to Evaluate Perception Models Using Planner-Centric Metrics, CVPR 2020) in a more general setting, and the problem discussed in this work is only a specific case. While this work focuses more on the adversarial attack, the core problems remain quite close: how to measure the impact of perception error from the system perspective (as once the proper metric is defined, one can always optimize either the attack for the bad and the perception module for the good).

That being said, the design of the proposed attack methods adopt a specific metric (traffic rule violation rate) on a particular AD system for optimisation. While the observation suggests the viability, the result may not be translated to broader impact trivially. E.g., besides the stop sign violation, how are other AD system metrics affected (hard brake, jerks, etc)? What if some modules of the AD system change (e.g., tracker, planner, etc)? What about other types of object hiding attack (e.g., pedestrians, vehicles, obstacles, lanes, etc)? There are various interesting cases that do not get studied and tested. Also, the experiment involves many random trials, yet the statistical significance is missing and it remains unknown how reliable these results are.

The presentation also needs to improve.
There are places where sentences do not make sense. E.g., “For example, the millions of Tesla cars (Kane, 2021) that are equipped with Autopilot (Tesla, 2022).”
The contents can be better organized. Critical results supporting the claim should be placed in the main text instead of appendix (e.g., table 4 and 5, figure 5, etc), in exchange of some implementation details.
Space between figure, table and text seem to be shrinked manually and affect readability (e.g., figure 2 and table 2).


**Strength And Weaknesses:**

Strength
Study on an interesting problem of practical significance

Weakness
Limited novelty
Missing literature and comparison to related works
Scenarios studied are too specific with limited impact and coverage.
Presentation should be improved


**Summary Of The Paper:**

The paper studies how a specific type of errors in the perception module of autonomous driving (AD) systems, when attacked adversarially, can be translated to the final consequence at the system level. The work empirically studies several popular adversarial attack strategies on stop sign detectors based on deep neural networks in a closed loop AD system simulator, and found that these approaches can not lead to traffic rule violation at all under the scenario considered. Two improvements that can enhance the impact of adversarial attacks are proposed: one being a more practical source distribution of the size of stop signs, and another one being the optimal range of attack to maximize the impact. Empirical study suggests that the proposed attack methods can lead to more serious consequences to the AD system.


**Summary Of The Review:**

An empirical study on adversarial attacks on autonomous systems with solutions limited only to a particular problem setting.

---

> ### Author Response · Authors · 2022-11-17
> **Authors' Response (Part 1)**
>
> Thank you for you comments and suggestions!
>
> **Reviewer 4: Novelty: On the novelty side, the impact of perception errors in the context of AD system has been studied by several recent works (e.g., Learning to Evaluate Perception Models Using Planner-Centric Metrics, CVPR 2020) in a more general setting, and the problem discussed in this work is only a specific case. While this work focuses more on the adversarial attack, the core problems remain quite close: how to measure the impact of perception error from the system perspective (as once the proper metric is defined, one can always optimize either the attack for the bad and the perception module for the good).**
>
> Thanks for the comments! First, we would like to clarify that we never intend to claim that how to (i.e, the *evaluation metrics*) measure the impact of perception error from the system perceptive is our contribution. Instead, we regard our main novel research contributions as (1) new measurement efforts of the AD system-level attack effects for existing adversarial AI attacks targeting AD. To the best of our knowledge, no such prior work has performed such system-level evaluations, i.e., with the full AD system pipeline and closed-loop control; (2) newly-discovered scientific insights from such new measurement efforts that quantitatively show that representative prior works actually cannot achieve their assumed AD system-level attack effect at all, which thus substantially differ from existing knowledge from prior works; and (3) Newly-identified and validated cause of such a scientific gap (H1, H2) and novel attack design methodology-level contribution to systematically improve system-level impacts. Thus, our contributions are orthogonal to the direct contribution (i.e., new evaluation metric design) of the CVPR ’20 paper “Learning to Evaluate Perception Models Using Planner-Centric Metrics”.
>
> Second, regarding the use of their metric design to address the large gap between perception errors and system-level attack effect, we would like to point out that no prior works have ever studied such potential usage before (e.g., the CVPR ’20 work did not even mention the adversarial AI attack context, not to mention the use of the metric to improve attacks), and also there exists various non-trivial applicability gaps. For example, their trained planner for downstream task approximation does not even consider STOP sign violation, making it not directly usable for guiding the generation of STOP sign violation-targeted adversarial AI attacks. In general, such a downstream task approximation is overly simplified if used to approximate real-world AD system behaviors, which are multi-threaded, with a mix of DNN and code, and with feedback loop, which is highly-difficult, if not impossible to use one single machine learning model to effectively approximate. After all, fundamentally our solution strategy is also orthogonal: instead of trying to effectively model the downstream tasks of real-world AD systems, we model the general physical properties of the system and operation scenario (e.g., braking distance, object size distribution with regard to a driving speed, etc.), which is agnostic to any specific AD downstream task design (since any such designs need to follow such basic physical properties) and thus can have the most general methodology effectiveness for any targeted AD system models.
>
> **[Revision]**
>
> We added an explicit clarification about our research novelty (with comparison to this CVPR work) in the introduction. The changes are marked as blue.

---

> ### Author Response · Authors · 2022-11-17
> **Authors' Response (Part 2)**
>
> **Reviewer 4: Methodology-level generality: that being said, the design of the proposed attack methods adopt a specific metric (traffic rule violation rate) on a particular AD system for optimisation. While the observation suggests the viability, the result may not be translated to broader impact trivially. E.g., besides the stop sign violation, how are other AD system metrics affected (hard brake, jerks, etc)? What if some modules of the AD system change (e.g., tracker, planner, etc)? What about other types of object hiding attack (e.g., pedestrians, vehicles, obstacles, lanes, etc)? There are various interesting cases that do not get studied and tested.**
>
> Thanks for the valuable feedback. We would like to point out that since H1 and H2 hypotheses  are both obtained from a general system model instead of specific to STOP-sign hiding attacks  (introduced in the Sec 2), our solution methodology based on them is actually generalizable at the design level to other system-level metrics, downstream algorithms, and victim object types beyond beyond the specific application scenario we evaluate on (i.e., STOP sign-hiding adversarial attacks for triggering STOP sign violations). For example, the designs for H1 and H2 are directly generalizable to object hiding attacks on road obstacles such as vehicles, pedestrians, and cyclists since the object size distribution based on the distance and the system-critical range (due to brake distance) still exist. In this case, the system-level metric is generalized to road obstacle collision beyond STOP sign violation, while the victim object type is generalized to vehicles, pedestrians, and cyclists beyond STOP sign. Also, since H1 and H2 are derived based on physical properties of the AD system and general operation scenario, the attack design improvements for them can be generally effective regardless of the AD downstream algorithm changes (since any such algorithms need to all follow such basic physical properties such as braking distance and object size distribution with regard to a driving speed).
>
> Thus, although just like any other single paper we can only afford to perform limited concrete prototyping and evaluation, our solution methodology design is actually generalizable and not limited to the specific problem setting we evaluate on. We will add explicit discussions on this.
>
> **[Revision]**
>
> We added explicit discussions on the solution methodology generality in Section 6.
>
> **Reviewer 4: Statistical significance of the results: the experiment involves many random trials, yet the statistical significance is missing and it remains unknown how reliable these results are.**
>
> |         | RP2     | RP_2 | RP_2 | RP_2| FTE-Y3| FTE-Y3 | FTE-Y3 | FTE-Y3 |  FTE-Y5| FTE-Y5 | FTE-Y5 | FTE-Y5 |
> |---------|------|---------------|-----|-------|------|---------------|-----|-------|------|---------------|-----|-------|
> |         | None | H1            | H2  | H1+H2 | None | H1            | H2  | H1+H2 |None | H1            | H2  | H1+H2 |
> | P-Value | -    | $4.4\times10^{-8}$ | 0.0    |     0.0  | -    | Nan | 0.08    |     $7.2\times10^{-8}$   | - | 0.08 | 0.04 | $1.4\times10^{-8}$|
>
>
> Thanks for the comments. To quantify the statistical significance, we calculate the P-Value for all trials in Table 5 compared with the None column. As shown, the P-Value is generally at the statistically significant level (e.g., generally <0.05 or at a similar magnitude, especially for the most important H1+H2 cases).
>
> **[Revision]**
>
> We have added the P-values in Table 5 and added related discussions in Section 5.1, which have been marked as blue.
>
> **Reviewer 4: Presentation need to improve: there are places where sentences do not make sense. E.g., “For example, the millions of Tesla cars (Kane, 2021) that are equipped with Autopilot (Tesla, 2022).” The contents can be better organized. Critical results supporting the claim should be placed in the main text instead of appendix (e.g., table 4 and 5, figure 5, etc), in exchange of some implementation details. Space between figure, table and text seem to be shrinked manually and affect readability (e.g., figure 2 and table 2).**
>
> **[Revision]**
>
> Thanks for the comments! We improved the paper based on your suggestions (e.g., relax the space) and also moved Table 4 and 5, Figure 5 into the main text in our revision.

---

> ### Author Response · Authors · 2022-11-22
> **Thank you for your helpful review! We hope you can check out our new results and our response/revision.**
>
> Dear Reviewer A81f,
>
> We are very grateful for your constructive feedback! Since the discussion period is closing soon, we sincerely hope you can have a chance to take a look at our response/revision and the new experimental results.
>
> In our response/revision, we include a clarification about our research novelty (especially compared with the CVPR paper you mentioned in your previous comments) in the Introduction section and a discussion about the generality indicating that our solution methodology design is actually generalizable and not limited to the specific problem setting we evaluate on (in paper Section 6). We also include P-Value experiment results to demonstrate the statistical significance of the results. Additionally, we improved the presentation and structure of our paper based on your suggestions. We hope that the reviewer can read our response and please kindly let us know if you have any additional comments or questions.
>
> Thank you!
>
> Sincerely,
>
> Anonymous Authors

---

> ### Comment · Reviewer_A81f · 2022-12-05
> **Response to the post-review responses from the author(s)**
>
> I have gone through the revision and responses. While some issues raised in my original review were fixed (e.g. statistical significance of the results), the major concern is still not addressed.
>
> The main take-away messages / contributions claimed in the submission are two-fold:
> - Effects of attacks or errors on an AD component evaluated by component-level metrics may not be translated to system-level consequences, and system-level metrics should be adopted for this purpose.
> - For the specific problem of stop sign hiding attack, previous baselines do not lead to AD stop sign violation, while improved strategies tailored for these attack methods do.
>
> The first one is hardly novel, as suggested in my initial comments (with reference to the the CVPR 2020 paper), by reviewer 3J9o, and even implied in the literature review of the submission itself, e.g., “AI component-level errors do not necessarily lead to system-level effects (e.g., vehicle collisions) (Dreossi et al., 2019; Seshia et al., 2022; Jia et al., 2020).”. As the result, in the specific case of stop sign hiding attack, evaluating the effect from the system-level perspective (the “new measurement efforts”, i.e., “to check if the AV under attack actually moves over the stop sign“) is a natural corollary and routine practice in this regard.
>
> The second one is implemented by explicitly exploiting the properties of specific adversarial attack methods (e.g., those with EoT for sampling the object size), within particular AD system-components in a simulation platform (e.g., it is assumed that there is an explicit tracking process with particular target deletion mechanisms in the perception module, which may not be the case in all AD systems), for a very concrete scenario (stop sign miss detection). With these assumptions, the applicable impact is restricted and transferability to general settings is unclear.
> I can see values in the engineering effort to fix the pitfalls in the baseline methods, but I'm not quite sure how well the incremental improvement is aligned with the prioritised focuses of ICLR.
>
> In summary, I kept my initial quantitative assessment unchanged.
>
> As a side note, there are still quite some presentation-wise issues in the revision that did not get addressed, to name a few (which can be easily spotted by a quick pass):
> - “This thus inevitably raise a critical research question” => should be “raises”?
> - “the system-level effects can be further improved, i.e., the violation rate can increase around 70%.” => “i.e.” should be “e.g.”?
> - “We evaluate our attack improvement in the platform we designed and show that the system-level effect can be significant improved” => “significant” should be “significantly”?
> - “The necessary first step is to perform a measurement study on the existing works about their system-level effects” => should be “The first necessary step”?
> - “Although in the majority of cases, H1 only cannot significantly improve the system-level effects (20% on average), …”  => what does “only cannot” mean?
>
> Most of these actually can be picked up by a trivial spell check. Again, a serious proofreading is really needed to make the current presentation less difficult to follow.

---

> > ### Author Response · Authors · 2022-12-13
> > **Authors' Response**
> >
> > Thanks a lot for your response! For the first part, although there exists some works, to the best of our knowledge, none of them (1) quantified such gaps in the context of adversarial attacks on autonomous systems, especially those in real-world system setups; and (2) identified novel designs that can systematically address/fill such gaps on autonomous systems. Without such a study, it is unclear whether the existing conclusions/claims are held or not. Thus we believe  that these two points are our novel and unique contributions.
> >
> > As for the focus of ICLR, in our discussion with Reviewer 3J9o, there are various such examples that were successfully accepted in the past [1-3], which have similar contributions as ours. Thus, we believe that our topic fits well in ICLR and we hope that the reviewer could be more objective when judging the fitness of our paper.
> >
> > For the presentation-wise issues, we thank the reviewers for their feedback. We will improve the presentation and perform serious proofreading for the next version of our paper.
> >
> > [1] Fooling Detection Alone is Not Enough: Adversarial Attack against Multiple Object Tracking, ICLR 2020
> >
> > [2] LowKey: Leveraging Adversarial Attacks to Protect Social Media Users from Facial Recognition, ICLR 2021
> >
> > [3] Characterizing Audio Adversarial Examples Using Temporal Dependency, ICLR 2019

---

### Official Review · Reviewer_VfwY · 2022-10-25

**Confidence:** 3
**Correctness:** 3
**Technical Novelty And Significance:** 3
**Empirical Novelty And Significance:** 3
**Recommendation:** 6

**Clarity, Quality, Novelty And Reproducibility:**

Clarity: the paper is written in a clear manner, with the goal clearly presented. However, some results presentation are not clear enough. For example, in table 3, it would be great if the caption of the table could contain more information such as the conclusion we can draw from the table, and what are the numbers about, and what are the differences between different testing cases.

Quality and novelty: the work is of relatively high quality if we consider the simulation to be authentic. The idea is novel as far as I know. It could be great if the author can provide some metrics showing the validity of the simulation platform and how quantitatively it is similar to the real world.

Reproducibility: the code is not released, and there are not enough training details such as network architecture, training hyper parameters. Therefore, the reproducibility is a question.

**Strength And Weaknesses:**

Strength: this is a very thought-provoking work. Many adversarial attack work have unreasonable or unrealistic assumptions, and may not work in the real world scenario. This work puts additional question marks on existing works and evaluate under real system constraints. The motivation and contribution of this work is relatively significant. The used method in this work also makes sense, and the used tools are relatively more realistic than other previous works.

Weakness: the experiments are still performed in simulation, and test is also done in simulation, which makes the result heavily rely on the validity of the simulator. However, it is not clear from the work how realistic is the used SVL simulator and if there is any convincing benchmarks or metrics that can validate the simulator.

**Summary Of The Paper:**

This work presents an interesting study of the effectiveness of existing object-hiding attack in autonomous driving from a system-level effects perspective. The authors proposed that the limitation of previous attacks is that they can only achieve successful attack for a particular targeted model such as object detectors, but can not achieve system level attack effects considering not only the perception module but also the control and vehicle physical properties. To be more practical, the authors proposed two limitation hypothesis that previous works have: the stop sign size in pixel sampled distribution is not uniform in the attack's system model when the vehicle is moving towards the stop sign; and prior works generate the attack without considering the system-critical range systematically, which leads to low performance of the attacks. The authors then proposed improvements on the system model in the AD context. The improvements include a new transformation distribution, and using a reasonable range to generate the attack. The results indicate that the proposed model is effective in achieving an average of 70% system violate rate.

**Summary Of The Review:**

Overall the work's idea is great and novel, and the achieved results outperform the baselines by a large margin, and it proposes to test the effectiveness of other existing work on attack on AD considering the system level constraints, which is of practical meaning.

---

> ### Author Response · Authors · 2022-11-17
> **Authors' Response**
>
> Thanks for the comments and suggestions!
>
> **Reviewer 3: Simulation fidelity: the experiments are still performed in simulation, and test is also done in simulation, which makes the result heavily rely on the validity of the simulator. However, it is not clear from the work how realistic is the used SVL simulator and if there is any convincing benchmarks or metrics that can validate the simulator.**
>
> Thanks for the comments!
>
> Note that for AD technologies, simulation-based evaluation has been not only widely adopted in related research in academia [5,6],  but also a standard methodology for testing AD systems in industry today due to the inherent limitations of real-road AD testing in cost, safety, efficiency, and corner-case coverage. For example, Waymo has reported more than 15 billion miles of such virtual testing of their AD system in simulation [1]; Aurora claimed that simulation-based testing enables them to perform 2.27 million unprotected left turns in simulation before even attempting one in the real world, which is the key to develop and deploy their AD system safely, quickly, and broadly [2]. Thus, we believe that using simulation for system-level evaluation is on par with the best and validated practices in both academia and industry.
>
> Regarding the specific simulator that we use, SVL, it is actually an production-grade high-fidelity AD simulator designed specifically for evaluating production-level AD systems [3]. It leverages Unity's built-in physics engine to accurately simulate the vehicle dynamics and tire-road interaction, and provide photo-realistic simulation of the driving environment that closely matches the real world [4]. It has already been demonstrated to be able to support production-grade systems [9]. As repeatedly demonstrated in various prior works, the end-to-end AD system-level evaluation results in SVL, especially those for adversarial AI attacks, can highly correlate with the same setup tested in the physical world [5, 6]. In our paper, to even further ensure the fidelity of our evaluation results, we further improved the fidelity of the rendering process by modeling the perception results in the real world with a practical setup (Section 3.2). Similarly, such high simulation fidelity has also been justified multiplied times for the control process. For instance, a research team at UC Berkeley has tested several representative scenarios generated in SVL [7] in a physical vehicle testing track, and concluded that SVL is “effective at synthesizing test cases that transfer well to the track” [8]. For instance, in [8] Fig. 6, autonomous vehicle and pedestrian trajectories obtained in simulation and real-world testing for the same test were qualitatively similar.
>
> **[Revision]**
>
> We include more details in the Section 3.2 which is marked as blue. Please refer to the newest version for the changes.
>
> [1] Waymo Safety Report, https://storage.googleapis.com/waymo-uploads/files/documents/safety/2021-12-waymo-safety-report.pdf, 2021
>
> [2] Scaling Simulation, https://aurora.tech/blog/scaling-simulation/, 2021
>
> [3] SVL Simulator: An Autonomous Vehicle Simulator, https://github.com/lgsvl/simulator/, 2022
>
> [4] LGSVL Simulator: A High Fidelity Simulator for Autonomous Driving, ITSC 2020
>
> [5] Too Afraid to Drive: Systematic Discovery of Semantic DoS Vulnerability in Autonomous Driving Planning under Physical-World Attacks,
> NDSS 2022
>
> [6] Dirty Road Can Attack: Security of Deep Learning based Automated Lane Centering under Physical-World Attack, USENIX Security 2020
>
> [7] Scenic: A Language for Scenario Specification and Scene Generation, PLDI 2019
>
> [8] Formal scenario-based testing of autonomous vehicles: From simulation to the real world, ITCS 2020
>
> [9] LGSVL, https://www.svlsimulator.com/about/, 2022
>
>
> **Reviewer 3: Clarity: the paper is written in a clear manner, with the goal clearly presented. However, some results presentation are not clear enough. For example, in table 3, it would be great if the caption of the table could contain more information such as the conclusion we can draw from the table, and what are the numbers about, and what are the differences between different testing cases.**
>
> **[Revision]**
>
> Thanks a lot for your suggestions! We improved the table captions (especially table 3, now it is table 5) to make it more clear, and the changes are marked blue.

---

> > ### Comment · Reviewer_VfwY · 2022-12-05
> > **Response to the rebuttal from the authors**
> >
> > Thanks for your rebuttal! My major concern is still with the simulation fidelity. From the images shown on the LGSVL simulator website, it seems there is still a large gap between the visual quality of scenes rendered by the simulator and the real world. For example, the texture is not photorealistic, and the buildings and other surrounding environments are not photorealistic, even if we are not talking about the effects of different light strengths and weather conditions. To evaluate that the attack's effectiveness, ideally these environmental factors should also be considered within your simulation.
> >
> > Overall I do value the motivation of the work to evaluate attack from a system level, but the experiments could be improved to show that it can really work in the real world.

---

> > > ### Author Response · Authors · 2022-12-13
> > > **Authors' Response**
> > >
> > > Thanks a lot for your response! Your criticism of our evaluation setup from the aspect of simulator visual fidelity is actually a **factual error**. In our revision and previous response, we explicitly point out that “to even further ensure the fidelity of our evaluation results, we further improved the fidelity of the rendering process by *modeling the perception results in the real world with a practical setup* (Section 3.2)”. That means all the perception results used in the simulation are **captured from the real world** instead of using the rendering image in the simulation which may lack fidelity. Thus, our simulation-based evaluation setup does not suffer from the simulator visual fidelity problem the reviewer pointed out. We hope that the reviewer can more correctly judge our work from this aspect.

---

### Official Review · Reviewer_fwRY · 2022-10-25

**Confidence:** 3
**Correctness:** 3
**Technical Novelty And Significance:** 3
**Empirical Novelty And Significance:** 3
**Recommendation:** 5

**Clarity, Quality, Novelty And Reproducibility:**

Clarity:
The logic of the paper is clear, from the formulation of the problem, to the establishment of the model, and the verification of the hypothesis, it clearly expresses the verification of the effectiveness of the system.
Quality:
Authentic professional expression, fluent language.
Novelty:
The concept of system model is creatively proposed, and some existing works are evaluated from the system level, and reasonable assumptions are put forward.
Reproducibility:
Looking forward to the performance of the code after open source.

**Strength And Weaknesses:**

strengths of the paper
Targeted attacks are carried out on the existing work (YOLO, Faster RCNN), and reasonable assumptions are put
forward according to the performance, and a more reasonable attack model is designed from the system model
level to better overcome the two limitations to achieve System level effectiveness.
weaknesses of the paper
W,r,t Hypothesis 1, how to determine the distribution. y= 1/x^2 is used to calculate the distribution in eq(3). Is
that the possible reason to balance the attack success rate between far distance and near distance?

**Summary Of The Paper:**

This article describes how to design an attack model more reasonably to evaluate target detection algorithms, thereby improving AD security. The authors propose the concept of a system model, and by attacking the previous work, it is found that the existing works cannot respond accordingly to the attack. Therefore, two assumptions are made that lead to low validity of the system-level evaluation.
The main contribution of this paper is to creatively propose the concept of a system model to evaluate the effectiveness of the attack, and to reasonably propose the assumption of uneven pixel distribution of the attack object, and design a system model-driven attack to verify the hypothesis.

**Summary Of The Review:**

In terms of attack evaluation technique, this paper is novel. But I think this paper might be more suitable for CVPR or other vision-related conferences as there is little theoretical contribution.

---

> ### Author Response · Authors · 2022-11-17
> **Authors' Response**
>
> Thank you for you comments and suggestions!
>
> **Reviewer 2: Clarity about the Hypothesis 1 (H1): weaknesses of the paper W,r,t Hypothesis 1, how to determine the distribution. y= 1/x^2 is used to calculate the distribution in eq(3). Is that the possible reason to balance the attack success rate between far distance and near distance?**
>
> Thanks for the comments!
>
> Yes, this can indeed in some sense be viewed as a formally-derived formula for achieving a systematic balance of the trade-off between attack success rates at far and near distances with regard to their effectiveness on triggering system-level attack effects (i.e., automatically optimize for the “system-critical” distance range to attack as in Section 4). Specifically, the victim object size distribution we derived based on the system model for H1 can systematically ensure that the victim object size distribution that the adversarial attack is “trained” for is consistent with the distribution that attack will be “tested” for (i.e., testing the adversarial attack in the physical world), since with fixed attack strength, testing performance is best when the testing distribution is similar to the training one. Here, the victim object size distribution is directly related to the attack distance, and thus can indeed be viewed as systematically achieving sufficient success on both far and near distances. As shown in Table 3, such formally-derived distribution formulas can indeed systematically achieve such a balance at the component level with regard to improving system-level attack effect.
>
> To make our derivation of the distribution more clear, we would like to include following more details in the paper: the distribution used in eq (3) (F’ \propto 1/s^2), where F’ is the number of frames and s is the STOP sign size in pixels, indicates the relationship between STOP sign size in pixels and the number of the frames of it (i.e., the frequency in the whole trip). Thus, if we know the distribution of the STOP sign size in pixels, we can know the probability distribution of the STOP sign size in pixels (s) in the whole driving trace, and thus we can easily integrate it into eq (3). We also define it in a formal way:
>
> We define the $S$ as such a discrete distribution, whose domain is $\{s_1, s_2, ..., s_N\}$, where $s_i$ is a STOP sign size in pixels. Based on the Eq (2), the probability of $s_i$ is $p(s_i) = \frac{1}{s_i^2}/\sum_{k = 1}^{N} \frac{1}{s_k^2}$.
>
> When applying the derived distribution formula to our evaluation settings, in order to get the STOP sign size in pixels, we use a camera-based rendering technique similar to prior work and leverage the nuScenes dataset (details are described in section 4.3) to render the STOP sign in the image. With the APIs provided by nuScences dataset, we can obtain the STOP sign size in pixels and then with its probability distribution, we can easily sample the STOP sign size with this distribution.
>
> **[Revision]**
>
> We include more details about the distribution like above in the revision in Section 4.3 with blue color.

---

### Official Review · Reviewer_3J9o · 2022-10-29

**Confidence:** 4
**Correctness:** 4
**Technical Novelty And Significance:** 2
**Empirical Novelty And Significance:** 2
**Recommendation:** 5

**Clarity, Quality, Novelty And Reproducibility:**

Clarity: Very Good
Quality:  Good
Novelty:  Marginal
Reproducibility:  The authors have reimplemented attack methods and offer to open source them. There is sufficient detail in the paper to be able to reproduce the experiments.

**Strength And Weaknesses:**

Strengths:
The main message of the paper is rather clear, the experimentation is done well, the link from real world to simulated data scenarios is nicely done and the results are conclusive.

Weaknesses:
While I appreciate the extensive systems engineering work, the novelty is marginal.  There is a lot of redundancy in the writing in the paper in the early sections and one could shorten them (many points are repeated multiple times).  I am wondering whether the paper is more suited for a conference such as WACV or CVPR since there are no general takeaways besides the concrete application setting insight.  The idea of performance characterization at a systems level rather than components level is well known in computer vision literature since the 90's. (see for instance: haralick.org ).

**Summary Of The Paper:**

The main essence of the paper is to point out that existing methods for illustrating how adversarial attacks can affect autonomous driving illustrate the problem by mainly taking a component evaluation perspective rather than a complete end to end systems evaluation perspective. The paper illustrates, by integrating stop sign attack data with an end to end autonomous driving stack, that failures at a component level are inconsequential at a systems level.  The authors trace the phenomenon to the nature of the sampling distribution of scale for stop signs and the fact that the particular error situations are never triggered given that tracking compensates for the misdetections due to adversarial perturbations. The paper then illustrates that one can redesign the attacks, by taking into account the projection geometry in the context, thus making the violations at a systems level to increase significantly.


**Summary Of The Review:**

While I appreciate the amount of engineering work in the paper to demonstrate how component level errors may not propagate at a systems level, the novelty is marginal. The paper may be more suited for WACV.

---

> ### Author Response · Authors · 2022-11-17
> **Authors' Response (Part 1)**
>
> Thank you for you comments and suggestions!
>
> **Reviewer 1: Novelty is marginal. The idea of performance characterization at a systems level rather than components level is well known in computer vision literature since the 90's. (see for instance: haralick.org ).**
>
> Thanks for the comments on the novelty side. To clarify, we never intend to claim that our novelty contribution is on being the very first to perform system-level performance characterization of vision algorithm errors in general. Instead, we view our main novel research contributions lie in (1) first to reveal and quantify the component-system gap in adversarial AI and autonomous system contexts; and (2) novel designs to address this gap.
>
> Specifically, compared to the line of research from Prof. Haralick pointed out by the reviewer (https://haralick.org/, e.g., [1, 2]), we are novel at least from 2 general research contribution dimensions (RD):
>
> *(RD1) Novel measurement efforts and insights of the CPS system-level effect of AI component-level errors from at least 3 dimensions: (1) adversarial AI attack, (2) autonomous systems, and (3) real-world systems.*
>
> To the best of our knowledge (based on the papers with accessible PDFs from the website), Prof. Haralick’s related research works mainly studied the propagations of image errors in vision systems. In comparison, our work has at least the following research-level novelty deltas:
>
> - [Measurement Scope-Level Delta #1] Prof. Haralick’s research focuses on innocent image errors (e.g., natural noises), while our work focuses on **intentional adversarial AI attacks.**
>
> - [Measurement Scope-Level Delta  #2] Prof. Haralick’s research focuses error propagations in vision systems, while our work is about error propagations in **autonomous systems**, which is actually a much larger “system” scope that encloses vision systems (e.g., perception module is only part of the whole perception-planning-control workflow in an autonomous system). Due to such vast difference, in his work he can use a non-linear function $F$ to represent the “system” [2], while for us it becomes highly difficult, if not impossible (e.g., the industry-grade AD systems we focus on all adopt modular designs [3] that are multi-threaded, with a mix of DNN and code, and with feedback loop, which is highly-difficult, if not impossible to use one single non-linear function $F$ to effectively model).
>
> - [Measurement Scope-Level Delta  #3] Prof. Haralick’s research focuses on abstract vision algorithms [2], instead of **real-world systems** like ours. Prof. Haralick’s works actually advocate the need to further study more concrete vision tasks [1, 2] (e.g., in [2] it was mentioned that “We are currently investigating the use of these techniques for important vision tasks like camera calibration, 3D reconstruction, and motion estimation.”), but none of his works actually performed such efforts. Our work is exactly filling such a gap on an important vision-based system, autonomous driving.
>
> - [Measurement Methodology-Level Delta] Prof. Haralick’s research mainly focuses on analytic analysis of the image errors on abstract vision algorithm formulation (e.g., the non-linear function $F$ [2]), while ours is a **quantitative analysis** on real-world systems. The former cannot be directly applied to a real-world system due to the lack of such an abstract formulation for the real-world system we want to study; thus, our quantitative analysis is both novel and necessary if we want to study such error propagations in a real-world system, especially the highly complicated real-world AI-enabled autonomous systems.
>
> - [Measurement Insights-Level Delta #1] Due to the lack of quantitative analysis of such error propagations above, Prof. Haralick’s work cannot reveal **how large such a component-system gap is in practice**. Instead, our work is able to show that the gap between component-level error and system-level attack effect in the AD context can be especially large, to the level that the former does not correlate with the latter at all (i.e., successful attacks at the component level can lead to complete failure at the system level, see Table 2). This can suggest that the commonly-viewed research success in a certain research area (in our case, physical object-hiding adversarial attacks for AD) may not be practically meaningful/useful at all, which is a research area-level insight that cannot be directly derived by Prof. Haralick’s work.

---

> > ### Comment · Reviewer_3J9o · 2022-11-17
> > **re: clarifications**
> >
> > Thank you for the detailed feedback from your point of view.   My original statement was about the fact that 'end to end' systems level characterization has been studied since the 90's.  My feedback was to give you a starting point to the literature as Prof. Haralick was one of the early ones to do performance characterization and perturbation analysis of vision systems.  There were many others in the 90's - Prof(s). Tom Binford, Prof. Wolfgang Foerstener and others who performed systems identification on chains of non-linear transformations.
> >
> > Real-world systems performance modeling has been studied in the industry for quite some time in the late 90's and early 2000's. While these are not at the scale of autonomous driving, the overall concept of application priors, end to end systems characterization of vision and control has been studied before.   Please see: Greiffenhagen et al (Proceedings of the IEEE ( Volume: 89, Issue: 10, October 2001), they had papers in CVPR 2000, 2001 that addressed end to end performance and had design optimization - e.g. optimal camera placement for meeting specific end performance goals in the context of a dual camera surveillance system. Moreover, they illustrated how one could evolve designs and achieve reuse of performance analysis when application context settings change.  This work was done in the Industry.
> >
> > The main aspect that I got from your paper is that component evaluations are not enough and that one needs to consider end to end system models. In my humble opinion, this insight is rather natural for someone with control theoretic systems viewpoint.   It is clear that adversarial perturbations (as in the context of deep learning brittleness) is something new to be addressed in modern contexts.

---

> > > ### Author Response · Authors · 2022-11-19
> > > **Authors' Response (Part 1-1)**
> > >
> > > Thank you so much for sharing more details on this and providing all these suggested related works! It is indeed our oversight with regard to the lack of sufficient writing-level emphasis on the large body of prior works in various other domains in terms of end-to-end system performance characterization and optimization. Although our contributions are different at various levels (e.g., the semantic problem context (autonomous systems), the angle of adversarial AI, the innovative use of physical models of vehicle system and environment to optimize system-level performance), we do agree that the our contributions can be generally viewed as end-to-end system performance characterization and optimization, and thus should have a sufficient treatment of them in the writing.
> > >
> > > **[Revision]**
> > >
> > > We expanded our novelty clarification in the introduction (marked in Blue) to further emphasize the existence of prior works in various other domains in terms of end-to-end system performance characterization and optimization.

---

> ### Author Response · Authors · 2022-11-17
> **Authors' Response (Part 2)**
>
> - [Measurement Insights-Level Delta #2] Besides showing a quantitative measure of the gap, we further diagnose the causes and identify **2 attack design-level insights** that can improve the system-level attack effect and also are systematically validated. Prof. Haralick’s work only studied error propagations in the vision system, but did not study how the “generation process” of such errors (i.e., in our case, the adversarial patch generation) can impact such an error propagation process (e.g., making the error propagate better or worse), while our novel insights at the attack design level are exactly about the latter. Such a insight-level difference is mainly because Prof. Haralick’s research is about innocent image errors (e.g., natural noises), instead of malicious attacks that can be intentionally generated (i.e., Measurement Scope-Level Delta #1 above).
>
> *(RD2) Novel attack design methodology-level contribution to improve system-level impacts.*
>
> Besides our novel measurement efforts and findings above (RD1), we also made design methodology-level contributions (i.e., the new system model-driven attack designs in Section 4.3), which is another category of scientific contributions in general. Prof. Haralick’s work is mainly about modeling and analyzing the image errors with respect to vision systems, but to the best of our knowledge, **none of his works have design contributions in improving such error propagations**. In our work, we are the first to achieve this in the context of adversarial AI attacks against AD.
>
>
> **[Revision]**
>
> We added an explicit clarification about our research novelty (with comparison to Prof. Haralick’s work) as discussed above in the introduction. The changes are marked as blue.
>
> [1] Performance characterization in image analysis: thinning, a case in point, Pattern Recognition Letters, 1992
>
> [2] Error Propagation For Computer Vision Performance Characterization, CISST 1999
>
> [3] A Survey of Autonomous Driving: Common Practices and Emerging Technologies, IEEE access 2020

---

> > ### Comment · Reviewer_3J9o · 2022-11-17
> > **See my comment for your previous response...**
> >
> > While I acknowledge that your work has merit, the main concern I have is the lack of knowledge or awareness of past work in performance modeling.  See for instance:  Thacker et al, CVIU, 2007,  https://doi.org/10.1016/j.cviu.2007.04.006  .  Also, please see previous comment on Greiffenhagen et al that addresses design aspects of video analytic systems.

---

> > > ### Author Response · Authors · 2022-11-19
> > > **Authors' Response (Part 2-1)**
> > >
> > > Thanks a lot for your comments! We address your concerns including the revision in the previous rebuttal (Part 1-1).

---

> ### Author Response · Authors · 2022-11-17
> **Authors' Response (Part 3)**
>
> **Reviewer 1: The paper is more suited for WACV/CVPR since there are no general takeaways besides the concrete application setting insight**
>
> Thanks for the valuable comments! We would like to argue that our work in fact does have various general takeaways beyond the specific application scenario we evaluate on (i.e., STOP sign-hiding adversarial attack against AD), e.g., generalizable both with regard to general AI attacks in the AD context and with regard to more secure/robust model training for autonomous systems in general, detailed as follows:
>
> - From the measurement point of view, the general takeaway is that in AD systems, **the gap between the component-level and system-level attack effects can be much larger than what prior works thought.**
>     - In most prior works, they believed that the component-level attack can be easily transferred to the system-level in AD systems. For instance, SIB paper [4] claims that “our AEs could potentially cause serious problems for autonomous driving cars.” However, in our paper, we evaluated the SIB in our practical AD setup and found that it can actually only achieve 0% system-level violation rate. This thus directly contradicts their claims and since the STOP sign-hiding adversarial attack is so far the most popularly studied AI attack in AD context, this suggests that in autonomous systems, the gap between the component-level and system-level attack effects can be generally larger than commonly expected, e.g., can be even to the level that a successful attack at the component level is pretty much meaningless/useless at the system level. This clearly suggests the need for system-level evaluation like what we performed in Section 3, since without which it can be very hard to know how meaningful a proposed attack is from the end-to-end AD driving perspective.
>
> - From the new attack methodology design point of view, **our two validated hypotheses are both generalizable to improve the system-level attack effect in other AD attack settings**
>     - The H1 and H2 hypotheses are both generalizable since they are both obtained from a general system model instead of specific to STOP-sign hiding attacks  (introduced in the Sec 2). For instance, if considering a pedestrian-hiding attack (instead of STOP sign), due to the motion of the vehicle, the different distribution and the system-critical range both still exist, and thus systematically considering them in the attack generation can help to improve the system-level effect. Specifically, in the pedestrian case, we only need to change the STOP sign to the pedestrian and add additional pedestrian movement area into the system model. In addition, our H1 can also be generalized to other downstream behavior models, e.g., by replacing the speed parameter v in Eq (2) with any longitudinal trajectory profile of interest.
>
> -  From the general solution direction point of view, **our system model-driven designs also provide general takeaways from secure/robust model training perspective**
>     - To train the autonomous driving AI model to avoid undesirable behaviors at the system level, it is highly desired to first have such concrete attack examples that can be meaningful/effective at the system level. In our paper, we fill in this essential gap by improving the AD system-level effects of the adversarial AI attack generation process with novel systematic integration of system models. We argue that such a system model-driven AI attack design approach can lead to direct general takeaways from the secure/robust model training perspective, since  the system model-driven attack designs (e.g., those for H1 and H2) can be directly used in adversarial training processes to more effectively train models to be more secure/robust against the adversarial AI attacks that can cause end-to-end system-level impacts. Moreover, the system models may also inspire the integration of the system model into further model training process to generally improve model security, robustness, or even accuracy in any AI-enabled systems instead of just for AD.

---

> > ### Comment · Reviewer_3J9o · 2022-11-17
> > **Rationale for WACV recommendation**
> >
> > The rationale for my  recommendation of your paper for WACV is largely because I considered a large body of prior work on systems analysis and found your work as incremental (that were not referenced and had insights about design aspects at end to end system level).   I did see value in your current contribution but it is my humble opinion that specific details of your paper are context specific.

---

> > > ### Author Response · Authors · 2022-11-19
> > > **Authors' Response (Part 3-1)**
> > >
> > > Thanks for your thoughtful considerations and suggestions! If possible, we still hope to have the chance to share our research at ICLR since as we described above, our findings can indeed have generalizable insights to important subareas in the ICLR community (e.g., safety and applications in vision and robotics). At ICLR, there are various such examples that were successful in the past [1-3] (i.e, having generalizable insights from a specific application context such as AD, face recognition, and automatic speech recognition) . We would like to kindly point out that venue fitness can be a largely subjective opinion, and thus we hope that different opinions in this will not be the reason to reject our work for ICLR.
> > >
> > > [1] Fooling Detection Alone is Not Enough: Adversarial Attack against Multiple Object Tracking, ICLR 2020
> > >
> > > [2] LowKey: Leveraging Adversarial Attacks to Protect Social Media Users from Facial Recognition, ICLR 2021
> > >
> > > [3] Characterizing Audio Adversarial Examples Using Temporal Dependency, ICLR 2019

---

> > > > ### Comment · Reviewer_3J9o · 2022-12-06
> > > > **Final summary**
> > > >
> > > > I thank the authors for their efforts to address my concerns.  After reading their feedback and other reviewer comments,  I concur with the comments of reviewer A81f (both on the novelty of the paper and the specificity of your contributions to this application context).   I believe that my overall rating is justified.

---

> > > > > ### Author Response · Authors · 2022-12-13
> > > > > **Authors' Response**
> > > > >
> > > > > Thanks a lot for your summary! With our previous response and revision, we point out our novel and unique contributions and also discuss the general takeaways. We hope that the reviewer could judge our paper in a more objective manner. The reviewer could point out our errors in the rebuttal and the revision and we will improve the draft in the next version instead of judging it in a subjective way. It would be great that the reviewer could provide more details about the revision/questions after reading our response and revision. Thanks!

---

> ### Author Response · Authors · 2022-11-17
> **Authors' Response (Part 4)**
>
> In additional, we would like to note that based on the ICLR 2023 Call For Paper (https://iclr.cc/Conferences/2023/CallForPapers), ICLR 2023 includes topics such as **safety and applications in vision and robotics**, which is essentially what’s driving the research contributions in our paper. Specifically, targeting AD, a safety-critical application of vision and robotics, we (1) perform a novel system-level attack effective measurement study on representative existing works, (2) identify new research area-level insight that all such prior attacks are having a huge gap toward achieving the meaningful/useful system-level effect they all claims to achieve, and (3) further analyze the causes and propose novel designs that have been validated to effectively address such a huge gap, with generalizable design-level insights on both attack and defense sides (see above).
>
> Thus, overall, we believe that our research contributions are well within the scope of ICLR, and also do have various generalizable takeaways beyond the insights specific to the concrete application setting we evaluate on.
>
>
> **[Revision]**
>
> We add a dedicated discussion on the general takeaways of our research in the discussion section (Section 6). The changes are marked as blue.
>
> [4] Seeing isn't Believing: Towards More Robust Adversarial Attack Against Real World Object Detectors, ACM CCS 2019

---

> > ### Comment · Reviewer_3J9o · 2022-11-17
> > **Please see comments to previous comment...**
> >
> > Same comment as for previous rebuttal.

---

> > > ### Author Response · Authors · 2022-11-19
> > > **Authors' Response (Part 4-1)**
> > >
> > > Thank you very much! We address your concerns with the revision in the previous rebuttal (Part 3-1).

---

### Comment · Area_Chair_73E3 · 2022-11-24
**Reviewers - please discuss**

We are now approaching the end of the discussion period and so far only a few of you have engaged in discussion with the authors.
The authors both updated the paper and provided detailed responses to the reviews. Please reply, clarifying whether your concerns were addressed and if not, why not.
Also, if your concerns were indeed addressed either update your score or clearly state why you still believe the score is appropriate.

Many thanks for making this conference a success and for taking your role seriously.

AC

---

### Decision · Program_Chairs · 2023-01-20

**Decision:**

Reject

**Justification For Why Not Higher Score:**

1) Since the approach is similar to prior work, the novelty is quite limited, appropriate for a workshop paper. Submitting to a benchmark tracks paper might be another option.
2) It is unclear to what extend the findings are robust to the specific implementation of the simulation, rather than generalisable.  For example, it is entirely unclear if any of the findings would translate to the real world.

**Justification For Why Not Lower Score:**

NA

**Metareview: Summary, Strengths And Weaknesses:**

This paper was marked as a borderline submission and discussed between the AC and some of the reviewers.
Overall the consensus was that the paper suffers from two different issues:
1) Since the approach is similar to prior work, the novelty is quite limited, appropriate for a workshop paper. Submitting to a benchmark tracks paper might be another option.
2) It is unclear to what extend the findings are robust to the specific implementation of the simulation, rather than generalisable.  For example, it is entirely unclear if any of the findings would translate to the real world.

All things considered the paper would benefit from addressing the reviewers' concerns and going through another round of fresh reviews. An alternative option is to submit it as a datasets and benchmark tracks paper at NeurIPS 2023, which seems to be a better fit for this type of work

**Summary Of Ac-Reviewer Meeting:**

Overall the consensus was that the paper suffers from two different issues:
1) Since the approach is similar to prior work, the novelty is quite limited, appropriate for a workshop paper. Submitting to a benchmark tracks paper might be another option.
2) It is unclear to what extend the findings are robust to the specific implementation of the simulation, rather than generalisable.  For example, it is entirely unclear if any of the findings would translate to the real world.